# Social, health and economic impact of COVID-19: Healthy Ageing In Scotland (HAGIS) – a protocol for a mixed-methods study

Stella Arakelyan ,[1] Tamara Brown,[1] Louise McCabe,[1] Lesley McGregor,[2] David Comerford,[3] Alison Dawson,[1] David Bell,[4] Cristina Douglas,[1] John Houston,[3] Elaine Douglas [1]

¹Centre for Environment, Dementia and Ageing Research (CEDAR), Faculty of Social Sciences, University of Stirling, Stirling, UK
²Division of Psychology, Faculty of Natural Sciences, University of Stirling, Stirling, UK
³Stirling Management School, University of Stirling, Stirling, UK
⁴Division of Economics, University of Stirling, Stirling, UK

**Correspondence to**
Dr Elaine Douglas;
elaine.douglas@stir.ac.uk

## ABSTRACT

**Introduction** Public health responses to the COVID-19 pandemic have reaped adverse physical, psychological, social and economic effects, with older adults disproportionally affected. Psychological consequences of the pandemic include fear, worry and anxiety. COVID-19 fear may impact individuals' mitigation behaviours, influencing their willingness to (re)engage in health, social and economic behaviours. This study seeks (1) to develop a robust and evidence-based questionnaire to measure the prevalence of COVID-19 fear among older adults (aged ≥50) in Scotland and (2) to examine the impact of COVID-19 fear on the willingness of older adults to (re)engage across health, social and economic domains as society adjusts to the 'new normal' and inform policy and practice.

**Methods and analysis** This mixed-method study includes a large-scale multimodal survey, focus groups and interviews with older adults (aged ≥50) living in Scotland, and an email-based 'e-Delphi' consultation with professionals working with older adults. The COVID-19 fear scale was developed and validated using exploratory and confirmatory factor analyses. Survey data will be analysed using descriptive and inferential statistics. Thematic analysis will be used to analyse qualitative data. Survey and qualitative findings will be triangulated and used as the starting point for an 'e-Delphi' consensus consultation with expert stakeholders.

**Ethics and dissemination** Ethical approval has been obtained from the University of Stirling for multimodal survey development, fieldwork methodology and data management. Anonymised survey data will be deposited with the UK Data Service, with a link provided via the Gateway to Global Ageing. Qualitative data will be deposited with the University of Stirling online digital repository—DataSTORRE. A dedicated work package will oversee dissemination via a coproduced project website, conference presentations, rapid reports and national and international peer-reviewed journal articles. There is planned engagement with Scottish and UK policy makers to contribute to the UK government's COVID-19 recovery strategy.

### STRENGTHS AND LIMITATIONS OF THIS STUDY

⇒ The survey sample will be based on existing participants of Healthy Ageing In Scotland and Generation Scotland which has the advantage of enabling analyses across time periods before and during the pandemic.
⇒ The large-scale survey and qualitative findings will be triangulated to provide robust evidence on the COVID-19 health, social and economic effects on older adults.
⇒ The survey sample has an inherent disadvantage of pre-existing sampling bias.
⇒ Multimodal survey data collection is likely to introduce selection bias which needs to be corrected by adjusting for observable correlates of bias such as age, gender and level of educational attainment.

## INTRODUCTION

The COVID-19 pandemic, caused by a novel SARS-CoV-2 coronavirus, has brought unprecedented disruption to our lives. Millions of people worldwide have been affected by the virus. Of these, older adults and those with underlying health conditions, have experienced disproportionately greater adverse effects.[1] At the time of writing, the number of confirmed COVID-19 cases has surpassed 305 million, and the number of deaths—5.4 million.[2] In the UK, the rate of deaths among those aged ≥60 attributed to COVID-19 (ie, COVID-19 on the death certificate) is considerably higher compared with younger age groups.[3] Similar trends are reported in other countries.[2]

Early evidence suggesting that older adults are at higher risk of contracting COVID-19 and developing suboptimal outcomes has prompted stringent government regulations seeking to protect this population.[1] The UK government issued guidance (eg, stay-at-home orders, shielding and social distancing)

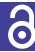

to safeguard vulnerable people during the COVID-19 pandemic. These measures, although necessary and effective in minimising the spread of the virus, had adverse mental health effects on older adults.[4] The pandemic has exacerbated feelings of social isolation and loneliness among older adults in the UK, with approximately one in two people aged ≥45 reporting feelings of loneliness.[5]

Increasing evidence demonstrates the detrimental impact of social isolation and loneliness on the physical health and well-being of the elderly (eg, increased blood pressure, heart disease, diminished immune system functioning, depression, anxiety, poorer cognitive functioning).[6] Older adults with disabilities and those living in areas affected by deprivation are particularly at risk for emotional distress, poor quality of life and low well-being.[4 7] The negative mental health and psychological well-being effects of the COVID-19 pandemic on older adults are well-evidenced in other countries.[8–10]

Fear is potentially mediating the link between the COVID-19 pandemic and its mental health and psychological well-being effects. Fear is characterised by emotive avoidance in relation to the stimulus.[11] Fear of encountering individuals who are possibly COVID-19 positive has been reported.[12] Evidence further suggests that behavioural responses to COVID-19 vary according to the level of COVID-19 fear.[13] A lower level of COVID-19 fear is associated with poor adherence to public health messages,[13] while an excessive level is associated with misattributing symptoms of seasonal colds or influenza as COVID-19 and poorer quality of life across physical and psychological health, social relationships and environmental domains.[14]

COVID-19 fear might affect individuals and society in a myriad of ways including (1) social isolation due to reservations around meeting others or engaging in pre-COVID-19 activities; (2) poorer health and well-being due to (A) reluctance to engage with health professionals for fear of contracting COVID-19 or (B) over-zealous self-referral due to high health anxiety and (3) weakened economic stability due to changing consumption and work patterns. There is a knowledge gap in relating these behavioural responses to COVID-19 fear among older adults, who are arguably most vulnerable to poorer outcomes from COVID-19.

### Research aims

The overall aim is to explore how the spectrum of COVID-19 fear manifests in older adults (aged ≥50) living in Scotland and how it impacts their health, social and economic behaviours.

### Research objectives

1. To develop a robust and evidence-based COVID-19 fear instrument to measure the prevalence of COVID-19 fear among older adults in Scotland.
2. To examine the impact of COVID-19 fear on the willingness of older adults to re-engage across health, social and economic domains as society adjusts to what

may be termed the 'new normal' and inform policy and practice.

### Main research questions

1. What is the prevalence of COVID-19 fear among older adults living in Scotland?
2. How has COVID-19 fear impacted social connectedness?
3. How and to what extent did older adults' use of information and communications technology (ICT) change during the lockdown?
4. What are the health predictors of COVID-19 fear, and are these socially graded?
5. How is COVID-19 fear associated with known barriers and facilitators to health service engagement (eg, general practitioner and hospital visits, screening programmes attendance)?
6. Has COVID-19 fear changed views about how long people expect to live and, if so, how far has it changed saving and spending behaviours?
7. How has COVID-19 fear changed consumption behaviours (eg, alcohol consumption, transport use, attendance at live events, dining-out, television viewing)?
8. How has COVID-19 fear affected views on workplace preferences and working patterns?

The findings will provide a robust understanding of how older adults in Scotland have negotiated their response to different aspects of their life in the 'new normal'. These insights, which are relevant to the UK as a whole, will inform policy and interventions on key social, health and economic issues pertinent to societal recovery from the COVID-19 pandemic.

### METHODS AND ANALYSIS
### Study design

This is a convergent, mixed-methods study comprising three phases: phase 1: development of validated COVID-19 fear scale (this phase was completed at the submission of this protocol for publication); phase 2A: a large-scale survey using multimodal data collection; phase 2B: individual and group interviews conducted by academic researchers and community-based coresearcher volunteers; phase 3: coproduction of findings with professionals working with older adults (e-Delphi exercise) to develop recommendations for policy and practice. A group of community-based coresearcher volunteers, sharing similar characteristics to our target population, has been guiding the development and implementation of these phases. Across the three phases, five work packages (WPs) are developed, reflective of the multifaceted nature of the aims and objectives of this study. The focus of each WP is presented in figure 1.

The quantitative strand of this work will promote the generalisability of findings in relation to the prevalence of COVID-19 fear among older adults and its impact on behavioural responses to the pandemic. The qualitative strand will help to develop a deep and rich understanding

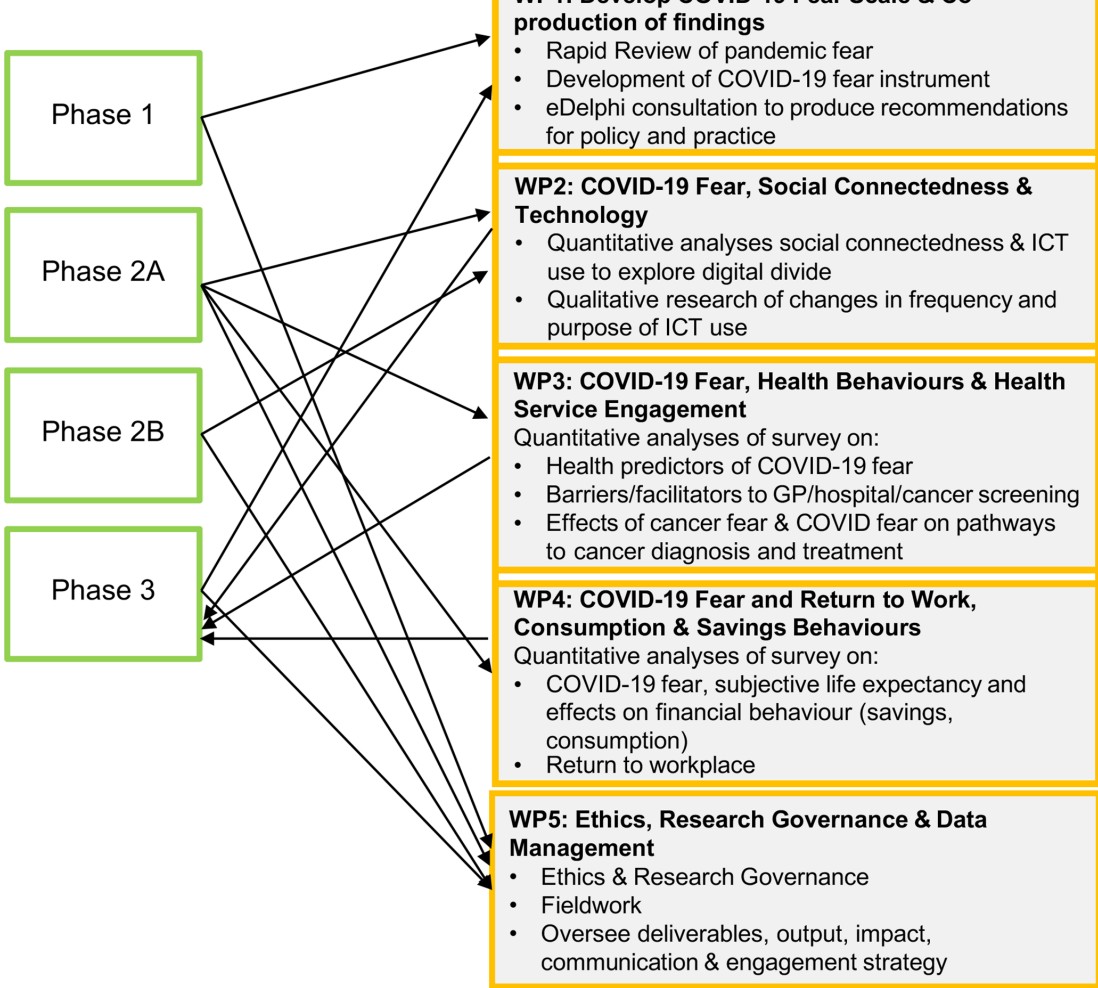

**Figure 1** Study phases linked to work packages.

of older adults' COVID-19 fears and worries, and pandemic-related experiences and behaviours. The consensus consultation with expert stakeholders will help to inform policy and practice, with a view to achieving impact for the study. The study dates are from December 2020 to November 2022.

### Phase 1: development of a scale to measure the prevalence of COVID-19 fear

#### Initial item development

We decided to develop and validate a new scale because the existing tools measuring COVID-19 stress, anxiety and/or fear were limited in scope, for example, were designed as clinical tools or focused on worries related to health and transmission. We required a scale that would measure a spectrum of fears and concerns in response to the pandemic more generally.

The initial development of the COVID-19 fear items was informed by a rapid review of scientific and grey literature. We searched databases available via EBSCOhost, Web of Science and ScienceDirect for English literature published since January 1991 and focus on (1) generic pandemic fear and health anxiety, and (2) COVID-19 fear, stress or

worry. After assessing results, we developed a protocol for a more focused rapid review (PROSPERO registration no: CRD42021250233). To meet review inclusion criteria, items needed to report the development or validation of instruments intended for use with adults to assess (1) the presence of a psychological state characterised as 'fear', 'worry', 'concern', 'anxiety' or other broadly synonymous descriptors, (2) the experience or measure of the psychological state which had been precipitated by awareness of or perceptions related to the COVID-19 pandemic and assessed in general or specific situations or in relation to specified contexts. We excluded literature describing (1) the development or validation of instruments in people aged ≤16; (2) not accessing the psychological states of interest; (3) combining assessment or measurement of fear with assessment or quantification of other personal characteristics (eg, personality traits, health conditions) and (4) assessing psychological states prompted by any other events, infectious agents or diseases. The rapid literature review was supplemented with a review of existing longitudinal studies that had COVID-19-specific modules (eg, COVIDLife, The Irish Longitudinal Study of Ageing,



English Longitudinal Study of Ageing). Based on this and literature review findings, the research team developed potential dimensions and drafted 100 candidate items.

### Sampling, recruitment and consent

For the item reduction exercise, we recruited participants using Prolific.co,[15] in two rounds. Prolific (formerly Prolific Academic) is an online platform on which academics post surveys for completion by a pool of participants. It has been demonstrated to produce high-quality data. In round 1, potential participants could see the survey advertised on Prolific if they were aged >18 with an IP address based within the UK. The study was advertised as a 5 min survey that asks 'about your attitudes to and perceptions of life in the UK these days' and included a warning that 'some of the questions ask about your experience of the pandemic and your worries'. In round 2, we used Prolific to filter potential respondents to those aged >40 with IP addresses based within the UK. The study was advertised as 'academic research on attitudes, expectations, and perceptions about the COVID-19 pandemic' and included a warning that the survey would cover 'topics of illness and death'. In both rounds of data collection, potential participants were offered payment for taking part, at a rate of £7.50 per hour. Potential participants were invited to view further details on the survey front page, which elicited their informed consent.

### Data collection and analysis

Data for Round 1 were collected between 12 March 2021 and 15 March 2021. A total of 262 respondents were recruited, of whom 241 completed all survey items. Explanatory factor analysis was conducted on the 100 candidate items using STATA V.15.1.[16] Explanatory factor analysis and parallel analysis identified candidate factors. Promax rotation identified items uniquely loading on each factor. These analyses delivered seven factors. The items associated with these factors were then tested for clarity and precision by coresearchers in our study in Think Aloud sessions.[17] The Think Aloud process allows for respondents to complete the survey in the presence of a researcher and speak their thoughts as they formulate their responses.

After checking the item comprehensibility with the coresearchers, we tweaked the wording of some items. We then recruited an additional 527 respondents for the second round of factor analysis. Participants were recruited using Prolific.co[15] between 4 May 2021 and 7 May 2021. We split the sample such that data from 263 respondents were used in exploratory factor analysis and data from the remaining 264 were used in confirmatory factor analysis.

The resultant multidimensional scale—the Worries Emerging from the COVID-19 Pandemic (WECP) scale—captures the following dimensions: worries about the future course of the COVID-19 pandemic; worries about readjusting to society; feelings of isolation; worries about the continuation or reintroduction of restrictions; worries

for family and friends; financial worries and worries regarding the safety and efficacy of COVID-19 vaccines. The WECP scale shows satisfactory internal consistency (as measured by Cronbach's alpha) as well as convergent and discriminant validity. The development, validation process and final scale are reported in the working paper by Comerford et al.[18]

## Phase 2A: a large-scale cross-sectional survey using multimodal data collection

### Sampling, recruitment and consent

Eligible participants will be older adults aged ≥50 who live in Scotland at the time of data collection. The target sample will be derived from two existing Scottish longitudinal studies—Healthy Ageing In Scotland (HAGIS) and Generation Scotland (n=15 074 participants) comprises. With an estimated response rate of 25%, this sample will potentially provide >3700 completed surveys. Response rates have varied over the period of the pandemic with some evidence of survey fatigue.[19] The respondents who previously consented to future recontact will be approached for consent to take part in this study. The mode of contact (online, postal and telephone) were based on prior expressed preference to support participation. Additionally, a predefined panel of 600 participants will be invited to take part in the online survey to address anticipated bias within the sample.

### Data collection and analysis

Online mode: Eligible online participants will receive an electronic invitation letter with an enclosed link to the study website (www.hagis.scot) and a personalised link to the survey. The website will describe the study, how to take part in the survey and how to get more information (including an email and a free phone number to connect directly to study staff). The online survey will take approximately 35–45 min to complete and will be hosted on the Qualtrics XM Platform.[20] Participants will receive a reminder following 2 weeks postinvitation.

Telephone mode: Participants for telephone interviews will be approached by DJS Research interviewers who will explain the study, how to get more information about the study and arrange a suitable time for the interview. The survey will take approximately 60–75 min to complete. Survey responses will be initially entered into the telephone-assisted personal interview system and then transferred into the Qualtrics XM Platform.[20]

Postal mode: All eligible postal participants will receive a postal invitation letter, information leaflet, paper-based survey and reply-paid envelope. The survey will take between 45 and 50 min to complete. All postal participants will be offered the option to take part in the survey electronically, in what may be termed as a 'nudge to web' approach. The reminder postcards will be sent to participants 3 weeks postinvitation.

Panel: DJS Research (social marketing agency appointed to support fieldwork activities) will recruit panellists to the study via an electronic invitation. The panellists who

express interest to participate will be directed to an online survey hosted by DJS Research using Nebu.[21] Panellists will be paid for completing the survey, at a rate of £12 per survey.

The survey will be largely based on the HAGIS survey[22] to include validated instruments for demographics, social circumstances, employment, physical health, mental health, health behaviour, social connectedness and social participation. This will be further developed by drawing on the research team expertise, enhanced by relevant literature searches to include other instruments, for example, vaccination status and attitudes. The WECP scale will be incorporated into the survey instrument. The survey instrument will be refined and pretested to ensure completion between 35 and 45 min, including the use of topic randomisation for the online mode. The survey was launched on 11 October 2021; survey fieldwork will be complete by end of January 2022.

Descriptive and inferential statistics will be generated using STATA V.15.1.[16] Differences between subgroups of continuous variables will be assessed by one-way analysis of variance and for categorical variables by the $\chi^2$ tests of independence. Correlates of COVID-19 fear and worries will be identified using univariate and multivariate regression analyses. Additional inferential statistical data analyses will depend on the specific research questions to be addressed in the health, social and economic WPs.

We should note that the survey sample has an inherent disadvantage of pre-existing sampling bias. We anticipate that there will be an over-representation of (1) older adults living in the East of Scotland, (2) those aged 55–65, (3) females and (4) those in the lower deciles of the income distribution. There will therefore likely be a concomitant under-representation of (1) older adults living in the South, West and North of Scotland, (2) the youngest and oldest sections of the older adult population, (3) males and (4) those at the lower ends of the income distribution. The sample weights should therefore will be estimated to align the survey participants as close as possible to the Registrar's General for Scotland's estimate of the structure of the older adult population in 2021. The sample weights will be calculated based on gender, locational and age-related imbalances and not the income distribution. Survey weights will be made available for analyses. Further, multimodal survey data collection is likely to introduce selection bias which needs to be corrected by adjusting for observable correlates of bias such as age, gender and level of educational attainment.

## Phase 2B: individual and group interviews
### Sampling, recruitment and consent
Semistructured individual and small group interviews (2–3 participants) will be conducted with 50 older adults (aged ≥50) who are currently living in Scotland and have the capacity to consent to participate in the study. Allocation to different activities will be according to the preference and availability of participants to support their engagement. Group interviews will enable participants to share their experiences with others, generating depth around shared experiences; individual interviews will enable a deeper exploration of individual experiences. Recruitment will be targeted to ensure representation of participants with diverse background characteristics—age category (50s, 60s, 70s and ≥80s), geographical location (rural and urban, across Scotland); gender, ethnicity, sexual orientation and socioeconomic position. A one-page recruitment advertisement poster will be designed to facilitate recruitment. The poster will be posted on the study's website and social media (Twitter, Facebook, Instagram) and the University of Stirling's website. Participants will also be recruited through the coresearchers' professional and social networks and the study partners' networks.

An information sheet will be provided to the potential participants; written informed consent will be sought prior to the interview being conducted. If the consent form is not returned before the interview, the member of the team will obtain oral consent, which will be recorded.

### Data collection and analysis
The semistructured interviews and focus groups will be conducted collaboratively by a member of the research team and a coresearcher. The format will be relatively flexible, with the technical aspects (ie, welcome and introduction, a brief presentation of the research, recording) being covered by the member of the academic team and the questions being asked by a coresearcher. For each interview and focus group, the researchers will meet approximately 30 min prior to the participant joining to discuss how the interview and focus group should be conducted. After the interview or focus group has concluded, the researchers will conduct a debrief discussion.

The interviews will take approximately an hour and focus groups around 1.5 hours. The participants will be asked whether they prefer to participate in an interview or a focus group. The interviews will be conducted either online, using the platforms Microsoft Teams or Zoom, or face to face (subject to individual preference and COVID-19 regulations). All focus groups will be conducted online for safety reasons. The topic guide includes questions on COVID-19 fear and worries, social and intergenerational connectedness, use of ICT (frequency and purpose) during the pandemic and digital exclusion. Data collection started in November 2021; the data collection is ongoing and expected to conclude in Spring 2022.

The data will be analysed in collaboration with the coresearchers using thematic analysis.[23] NVivo V.12[24] will be used for deductive coding of data.

## Phase 3: development of recommendations for policy and practice
### Sampling, recruitment and consent
An email-based e-Delphi consultation informed by Belton et al's [25] 'six-step prescription' will combine findings from qualitative and quantitative strands across

multiple iterations of asynchronous consultation with a panel of 30 professionals working with older adults across social, health and economic contexts. Relevant organisations (service providers, charitable organisations) will be asked to suggest a potential expert panellist who may join the study. Potential panellists will be sent an information sheet containing details of what will be involved in an e-Delphi consultation and asked to provide written consent to participation before the consultation begins.

### Data collection and development of recommendations

Panellists will complete three rounds of an electronic survey. The first will explore panellists' perceptions of changes in older adults' engagement with social, health and economic activities during the pandemic and share their perceptions of the needs and priorities for impact (eg, intervention and/or policy). The second survey will be informed by the summary analysis of the first survey, preliminary findings from preceding study fieldwork (phase 2) and will re-explore the panel's recommendations and priorities. The third and final survey, accompanied by a summary analysis of the second, will explore the extent to which, and reasons why panellists prioritise their recommendations. The output of the e-Delphi exercise will produce a set of recommendations for action that incorporate the rationale and priorities based on professionals' experience and expertise.

Through this iterative approach, we will arrive at recommendations for practice and policy to ameliorate the impacts of COVID-19 fear. The recommendations will take into account the experiences of older adults as reflected in the qualitative and quantitative data from other phases of the study and will be informed by expert panellists' real-world experience when supporting older adults during the pandemic.

### Patient and public involvement

We have established a group of up to twelve older adults (aged ≥50) living in Scotland who acted as community-based coresearcher volunteers within the study. The primary aim of this group is to ensure that the voices of adults over 50 are appropriately represented in the study and for them to act as 'experts by experience' across all study phases and related activities. Their input will be key in ensuring that the study participants are provided with research documentation that enables them to make fully informed choices around participation and consent, and that study outputs are accessible to a wide range of professional and lay stakeholder groups. The coresearchers will meet monthly using an online platform and in-person when possible, taking into account the Scottish Government and University guidance. Coresearchers will be provided with equipment and training to support their engagement online as well as research training to enable qualitative data collection and analysis.

Coresearchers have been recruited through existing networks of the research team as well as announcements on social media. Members of the team have provided

information and terms of reference for the group and asked to sign a volunteer agreement. The coresearchers will be actively engaged in the development of the COVID-19 fear scale and design of the survey questions; design of interview and focus groups topic guides; qualitative data collection; qualitative data analysis; interpretation of research findings; and communication of research outputs to a lay audience.

### Ethics and dissemination

We have a dedicated WP responsible for Ethics, Research Governance and Data Management (figure 1). The study seeks to recruit participants of the HAGIS pilot study and extend the sample to Generation Scotland participants. Recontacting these participants will require the processing of identifiable data, including contact details (postal address, telephone number, email). These data will be used for reconsent purposes and the provision of survey only. Returned survey questionnaires will use anonymised reference codes to protect the privacy and ensure anonymity. Data management will be guided by ESRC Research Data Policy.[26] Key ethical considerations will be (1) data security and anonymity; and (2) potential sensitivity of COVID-19 fear topic to the older age group. The research team is experienced in the handling of sensitive data and knowledgeable of protocols, best practices and ethical and legal requirements for processing this type of data. We will ensure the adherence and compliance of the research team to standard protocols and practise (ie, Data Process Impact Assessment, Data Sharing Agreements, the UK General Data Protection Regulation (UK GDPR)). All researchers involved in data collection will complete the MRC Research, UK GDPR and Confidentiality training courses. We will anticipate and plan for a potential upset that may be caused by sensitive topics. All study participants will be provided with a study email and a Freephone number to contact the team. Direct contact details to the Principal Investigator will be made available to all those approached to take part in the study.

Ethical approval has been obtained for the development of a validated COVID-19 Fear Scale, the establishment of the Older Persons Advisory Group (coresearchers), the survey development and fieldwork, conduct of interviews and focus groups from the General University Ethics Panel at the University of Stirling. To enable the rapid commencement of projects, the University of Stirling has brought in an expedited review of applications for ethics approval and priority response from legal, human resource and finance professional services. Data sharing with other parties will be subject to a Data Sharing Agreement, the use of strict security protocols and ethical approval.

The survey will provide a rich data resource accessible for future scientific research. Anonymised survey data will be deposited with the UK Data Service.[27] The data will be accessible free of charge for non-commercial users. We will conform to the Data Documentation Initiative standard, which is used by the UK Data Service. The link

to deposited data will be made available on the study website (www.hagis.scot) and via the Gateway to Global Aging[28]—a public platform developed to facilitate cross-national and longitudinal analyses of studies focusing on ageing, health and retirement around the world. Our study is included in the Gateway's digital library to facilitate national and international research. Qualitative data will be deposited with the University of Stirling online digital repository—DataSTORRE.

We will share the findings via the study's website, rapid reports, academic publications, webinars and presentations at national and international conferences. Rapid reports will provide timely access to emerging findings and academic publications to address key research questions. There is planned dissemination to Scottish and UK policymakers and partners. An Expert Advisory Board will be established to provide opportunities for the study to receive feedback and advice, and to consolidate relationships between the network of interdisciplinary experts in ageing studies, gerontology, economics and public health to support and sustain HAGIS in the longer term.

**Acknowledgements** We are thankful to Olivia Olivarius and Cate Pemble for their support with the development of the Worries Emerging from the COVID-19 Pandemic (WECP) scale and the survey instrument. We are extremely grateful to our community-based coresearchers Roy Anderson, Elizabeth Chrystall, David Curry, Margot Fairclough, Christine Ritchie, Pat Scrutton and Ann Smith who have contributed extensively to the development of project materials and qualitative fieldwork.

**Contributors** ED developed the research concept and design and gained funding. ED, SA, TB, DC, LoM, LeM, JH, DB, AD and CD contributed to the development of the first draft of the manuscript. SA revised and edited the subsequent manuscript drafts based on the comments from all the authors. All authors read and approved the final manuscript.

**Funding** This work was funded by the Economic and Social Research Council (ESRC) as part of the UK Research and Innovation (UKRI) rapid response to COVID-19. Grant number: ES/V01711X/1.

**Competing interests** None declared.

**Patient and public involvement** Patients and/or the public were involved in the design, or conduct, or reporting, or dissemination plans of this research. Refer to the Methods section for further details.

**Patient consent for publication** Not applicable.

**Provenance and peer review** Not commissioned; externally peer reviewed.

**ORCID iDs**
Stella Arakelyan http://orcid.org/0000-0003-0326-707X
Elaine Douglas http://orcid.org/0000-0001-8540-1126

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
