## [Reviewer comments · BMJ Open]

ARTICLE DETAILS

TITLE (PROVISIONAL)	The social, health and economic impact of COVID-19 – Healthy Ageing In Scotland (HAGIS): a protocol for a mixed-methods study
AUTHORS	Arakelyan, Stella; Brown, Tamara; McCabe, Louise; McGregor, Lesley; Comerford, David; Dawson, Alison; Bell, David; Douglas, Cristina; Houston, John; Douglas, Elaine

VERSION 1 – REVIEW

REVIEWER	Jansen, Tessa National Institute for Public Health and the Environment, Nutrition, Prevention and Care
REVIEW RETURNED	17-Mar-2022

GENERAL COMMENTS	The study protocol is sound and (for the greater part) clearly detailed. I do have some comments regarding the aim and problem statement of the study and some questions that I would like to be clarified. Introduction & problem statement: - Fear is central in the problem statement, however the rationale provided in the introduction does not convince me to support this narrow focus on barriers to re-engage in 'normal life' by older people. Are other factors sufficiently taken into account? In my opinion it would be more informative to study whether older people are willing to engage in normal life again and what facilitators and barriers they experience. So I would recommend to take up a broader view. The research questions are still valid if they are reformulated to cover a wider spectrum of feelings and perceptions.- I would like to see references to support the assumption that people may have poorer health outcomes due to reluctance to engage with healthcare providers in fear of contracting COVID-19. I believe that may have been the case in the early phases of the outbreak, but I doubt whether that could still be a factor. (Since the study was conducted in early 2021, this might have been a relevant factor, however this should be supported by a clear rationale based on previous literature). Methods & analysis: - The study is ongoing for quite a while and ends June 2022. Therefore, my comments regarding the study's emphasis may be addressed in reporting of the study's findings.- Please provide more details about the use of Prolific in recruiting participants.- Phase 2B: Interviews and focus groups: semi-structured interviews and small focus groups are conducted. Please detail how people were/will be matched for focus groups (for instance
---

	with similar/dissimilar characteristics to yield discussion or availability, preference, etc.) and why that particular approach was followed.  - The authors might want to consider to add experiences and response rates from previous studies using the healthy aging and generation panels. And, if applicable what was learnt from and improved after these previous studies. Are there any risks for low response rates for instance? Response enhancing strategies are detailed, however how are other possible risks tackled? - Please describe in more detail the use of weighting in the analysis. Minor:  - Past and present tense are used interchangeably.
--	---

REVIEWER	Shin, Su University of Utah Health
REVIEW RETURNED	06-Jul-2022

GENERAL COMMENTS	This manuscript describes a protocol for conducting important research addressing issues related to the current COVID-19 pandemic. I am looking forward to reading their final manuscripts laid out in this protocol, which will present the results. I have a couple of comments on this manuscript.  1. The importance of "fear." It seems that multiple studies laid out in this protocol use "COVID-19-related fear" as a main explanatory variable. The authors discussed why it matters in the introduction, but their justification appears to be still weak. It might be better to include more discussions on what they are trying to measure using their items. In their plan, is fear a multidimensional concept? If so, it might be useful to provide information about the exact dimensions of fear they try to measure. The reason to suggest this is that based on my research and literature review fear itself is not the main drive behind older adults' mental health issues. Instead, their own evaluation of the risk of infecting COVID-19 or dying from it, governmental health mandates and guidelines (e.g., lockdown, etc.), and economic insecurity caused by the governmental health mandates influence people's (including older adults') mental health. Therefore, more justifications about "why" fear matters more than other factors should be highlighted. 2. Justification of developing their own "fear" measure Developing their own fear measure is one of the study objectives. However, their justifications of "why" they are developing their measure are not clear. 3. Fear as a causal predictor of consumption behaviors, work status, etc. This protocol describes that the authors will examine a causal link between fear and outcomes. In empirical studies, identifying causal relationships is not an easy task. However, the protocol does not provide any information about how they will proceed with this task at all. If they cannot present the results based on causation, it would be better to use "correlations", not 'causations." Minor comments:
--

	1. Most studies using this age group use the term "older adults," not "older people." It would be better to replace older people with older adults.2. p4. Only strengths of this study are listed. Are there no expected limitations?
--	--

VERSION 1 – AUTHOR RESPONSE

Responses to Reviewers' Comments	
Reviewer 1 comment	Author response
Introduction & problem statement: Fear is central in the problem statement, however the rationale provided in the introduction does not convince me to support this narrow focus on barriers to re-engage in 'normal life' by older people. Are other factors sufficiently taken into account? In my opinion it would be more informative to study whether older people are willing to engage in normal life again and what facilitators and barriers they experience. So I would recommend to take up a broader view. The research questions are still valid if they are reformulated to cover a wider spectrum of feelings and perceptions.	Thank you for this pertinent comment. The study was formulated to address the extent to which fears and concerns about COVID-19 may affect social, health and economic behaviours as part of a comprehensive survey and mixed methods study. Overall, the study addressed many of the facilitators and barriers to a range of behaviours. The development of the Worries Emerging from the Covid Pandemic scale was designed to address concerns as a continuum from little/no concern to high concern. This scale can then be tested for an association with a wide range of behaviours.
I would like to see references to support the assumption that people may have poorer health outcomes due to reluctance to engage with healthcare providers in fear of contracting COVID-19. I believe that may have been the case in the early phases of the outbreak, but I doubt whether that could still be a factor. (Since the study was conducted in early 2021, this might have been a relevant factor, however this should be supported by a clear rationale based on previous literature).	Health engagement was one example of an area that we wished to explore. Our hypothesis was based on work that demonstrates that those with greater fear of cancer delay engagement with health services, e.g., see papers by: (1) Vrinten, C., McGregor, L. M., Heinrich, M., von Wagner, C., Waller, J., Wardle, J., & Black, G. B. (2017). What do people fear about cancer? A systematic review and meta-synthesis of cancer fears in the general population. Psycho-oncology, 26(8), 1070-1079; (2) Quaife, S. L., Waller, J., von Wagner, C., & Vrinten, C. (2019). Cancer worries and uptake of breast, cervical, and colorectal cancer screening: a population-based survey in England. Journal of medical screening, 26(1), 3-10. At the time of producing the protocol, little was known about the effects of COVID-19 fear and worries on health engagement. However, we wished to explore if it was a barrier to engagement.
Methods & analysis:	

The study is ongoing for quite a while and ends June 2022. Therefore, my comments regarding the study's emphasis may be addressed in reporting of the study's findings.	Thank you. Yes, results are emerging, and we are in the process of preparing other reports and papers for publication.
Please provide more details about the use of Prolific in recruiting participants.	We have now provided more details on participant recruitment via Prolific and further analysis. Please refer to pages 10-11 to see the changes which read: For the item reduction exercise, we recruited participants using Prolific.co¹⁵, in two rounds. Prolific (formerly Prolific Academic) is an online platform on which academics post surveys for completion by a pool of participants. It has been demonstrated to produce high quality data. In round one, potential participants could see the survey advertised on Prolific if they were aged >18 with an IP address based within the UK. The study was advertised as a 5 minute survey that asks "about your attitudes to and perceptions of life in the UK these days" and included a warning that "some of the questions ask about your experience of the Pandemic and your worries". In round two, we used Prolific to filter potential respondents to those aged >40 with IP addresses based within the UK. The study was advertised as "academic research on attitudes, expectations and perceptions about the COVID-19 pandemic" and included a warning that the survey would cover "topics of illness and death". In both rounds of data collection, potential participants were offered a payment for taking part, at a rate of £7.50 per hour. Potential participants were invited to view further details on the survey front page, which elicited their informed consent. After checking the item comprehensibility with the co-researchers, we tweaked the wording of some items. We then recruited an additional 527 respondents for the second round of factor analysis. Participants were recruited using Prolific.co¹⁵ between 4 and 7 May 2021. We split the sample such that data from 263 respondents were used in exploratory factor analysis and data from the remaining 264 were used in confirmatory factor analysis. The resultant multidimensional scale - the Worries Emerging from the COVID-19 Pandemic (WECP) scale - captures the following dimensions: worries about the future course of the COVID-19 pandemic; worries about readjusting to society; feelings of isolation; worries about the continuation or reintroduction of restrictions; worries for

	family and friends; financial worries and worries regarding the safety and efficacy of COVID-19 vaccines. The WECP scale shows satisfactory internal consistency (as measured by Cronbach's alpha) as well as convergent and discriminant validity. The development, validation process and final scale are reported in the working paper by Comerford et al., 2022.¹⁸
Phase 2B: Interviews and focus groups: semi-structured interviews and small focus groups are conducted. Please detail how people were/will be matched for focus groups (for instance with similar/dissimilar characteristics to yield discussion or availability, preference, etc.) and why that particular approach was followed.	We have now removed the term focus group and replaced it with 'small group interview'; this is to describe the planned activities more accurately. Further information is provided on the recruitment activity for and purpose of these two activities.
The authors might want to consider to add experiences and response rates from previous studies using the healthy aging and generation panels. And, if applicable what was learnt from and improved after these previous studies. Are there any risks for low response rates for instance? Response enhancing strategies are detailed, however how are other possible risks tackled?	We have added two sentences to reference the changes in response rates during the pandemic as experienced by an online only survey conducted by our partner, Generation Scotland. In our study, we adopted a mixed modal approach (based on participants prior expressed preference) and a 'nudge to web' offering to all postal participants. Please refer to page 11 to see changes: Response rates have varied over the period of the pandemic with some evidence of survey fatigue.¹⁹ The mode of contact (online, postal and telephone) were based on prior expressed preference to support participation.
Please describe in more detail the use of weighting in the analysis.	We have now expanded our descriptions on weighing. Please refer to page 13 to see the changes which read: We should not that the survey sample has an inherent disadvantage of pre-existing sampling bias. We anticipate that there will be an over-representation of (i) older adults living in the East of Scotland, (ii) those

	aged 55-65, (iii) females, and (iv) those in the lower deciles of the income distribution. There will therefore likely be a concomitant under-representation of (i) older adults living in the South, West and North of Scotland, (ii) the youngest and oldest sections of the older adult population, (iii) males, and (iv) those at the lower ends of the income distribution. The sample weights should therefore will be estimated to align the survey participants as close as possible to the Registrar's General for Scotland's estimate of the structure of the older adult population in 2021. The sample weights will be calculated based on gender, locational and age-related imbalances and not the income distribution. Survey weights will be made available for analyses.
Past and present tense are used interchangeably.	Thanks for your observation. We have now added a sentence to make it clear that phase 1 of this project was completed before submission of the protocol for publication. Please refer to page 8 to see changes which read: This is a convergent, mixed-methods study comprising three phases: Phase 1: development of validated COVID-19 fear scale (this phase was completed at the submission of this protocol for publication); Phase 2A: a large-scale survey using multimodal data collection; Phase 2B: individual and group interviews conducted by academic researchers and community-based co-researcher volunteers; Phase 3: co-production of findings with professionals working with older adults (e-Delphi exercise) to develop recommendations for policy and practice. We have also changed the present tense to future for clarity.
Reviewer 2 comments	
The importance of "fear." It seems that multiple studies laid out in this protocol use "COVID-19-related fear" as a main explanatory variable. The authors discussed why it matters in the introduction, but their justification appears to be still weak. It might be better to include more discussions on what they are trying to measure using their items. In their plan, is fear a	We have described the rationale for developing a new scale in our working paper by Comerford et al., 2022; this paper is also cited in our protocol. We have now added 2 paragraphs to explain the reasons for developing a new tool and describe the dimensions of the WECP scale. Please see pages 9 and 11 to see changes that read:

multidimensional concept? If so, it might be useful to provide information about the exact dimensions of fear they try to measure. The reason to suggest this is that based on my research and literature review fear itself is not the main drive behind older adults' mental health issues. Instead, their own evaluation of the risk of infecting COVID-19 or dying from it, governmental health mandates and guidelines (e.g., lockdown, etc.), and economic insecurity caused by the governmental health mandates influence people's (including older adults') mental health. Therefore, more justifications about "why" fear matters more than other factors should be highlighted.	We developed and validate a new scale because the existing tools measuring COVID-19 stress/anxiety and/or fear were limited in scope, e.g., were designed as clinical tools or focused on worries related to health and contamination. We required a scale that would measure a spectrum of fears and concerns in response to the pandemic more generally The resultant multidimensional scale - the Worries Emerging from the COVID-19 Pandemic (WECP) scale - captures the following dimensions: worries about the future course of the COVID-19 pandemic; worries about readjusting to society; feelings of isolation; worries about the continuation or reintroduction of restrictions; worries for family and friends; financial worries and worries regarding the safety and efficacy of COVID-19 vaccines. The WECP scale shows satisfactory internal consistency (as measured by Cronbach's alpha) as well as convergent and discriminant validity. The development, validation process and final scale are reported in the working paper by Comerford et al., 2022.¹⁸
Justification of developing their own "fear" measure Developing their own fear measure is one of the study objectives. However, their justifications of "why" they are developing their measure are not clear.	Please see the answer provided above.
Fear as a causal predictor of consumption behaviours, work status, etc. This protocol describes that the authors will examine a causal link between fear and outcomes. In empirical studies, identifying causal relationships is not an easy task. However, the protocol does not provide any information about how they will proceed with this task at all. If they cannot present the results based on causation, it would be better to use "correlations", not 'causations."	In our data analysis plan for Phase 2, we made it clear that our intentions are to identify correlates of COVID-19 fear and test associations. Please refer to page 13 to read our descriptions: Correlates of COVID-19 fear will be identified using univariate and multivariate regression analyses. Additional inferential statistical data analyses will depend on the specific research questions to be addressed in the health, social and economic work packages.

Most studies using this age group use the term "older adults," not "older people." It would be better to replace older people with older adults.	We have changed "older people" to "older adults" throughout the manuscript.
p4. Only strengths of this study are listed. Are there no expected limitations?	We have now addressed this; please refer to page 4: -The survey sample has an inherent disadvantage of pre-existing sampling bias. -Multimodal survey data collection is likely to introduce selection bias which needs to be corrected by adjusting for observable correlates of bias such as age, gender, and level of educational attainment. We have also expanded on this limitation on page 13: We should not that the survey sample has an inherent disadvantage of pre-existing sampling bias. We anticipate that there will be an over-representation of (i) older adults living in the East of Scotland, (ii) those aged 55-65, (iii) females, and (iv) those in the lower deciles of the income distribution. There will therefore likely be a concomitant under-representation of (i) older adults living in the South, West and North of Scotland, (ii) the youngest and oldest sections of the older adult population, (iii) males, and (iv) those at the lower ends of the income distribution. The sample weights should therefore will be estimated to align the survey participants as close as possible to the Registrar's General for Scotland's estimate of the structure of the older adult population in 2021. The sample weights will be calculated based on gender, locational and age-related imbalances and not the income distribution. Survey weights will be made available for analyses. Further, multimodal survey data collection is likely to introduce selection bias which needs to be corrected by adjusting for observable correlates of bias such as age, gender, and level of educational attainment.
Editor's comments	

Please revise the ‘Strengths and limitations of this study’ section of your manuscript (after the abstract). This section should contain up to five short bullet points, no longer than one sentence each, that relate specifically to the methods. The novelty, aims, anticipated results or expected impact of the study should not be summarised here.	We have revised this. Please refer to page 4 to see changes which read: -The survey sample will be based on existing participants of HAGIS and Generation Scotland which has the advantage of enabling analyses across time periods before and during the pandemic. -The large-scale survey and qualitative findings will be triangulated to provide robust evidence on the COVID-19 health, social and economic effects on older adults. -The survey sample has an inherent disadvantage of pre-existing sampling bias. -Multimodal survey data collection is likely to introduce selection bias which needs to be corrected by adjusting for observable correlates of bias such as age, gender, and level of educational attainment.
--	---

VERSION 2 – REVIEW

REVIEWER	Shin, Su University of Utah Health
REVIEW RETURNED	15-Oct-2022

GENERAL COMMENTS	I have minor comments to improve on the current manuscript. I have listed more important ones first. Major comments: 1. (Page 10) Lines 9-40: The authors describe their sampling method to develop the fear instrument. Their target sample of the entire project is adults aged 50 or older. However, the research used online panel 1) those who aged > 18 in the first stage and 2) those who aged > 40 in the second stage. Are these relevant? Do you have rationales for this age group? Also, it seems like the authors used online panelists who are used to participate in online surveys. Are they representative to the national sample? They must show that their sample characteristics are similar to those of people aged 50 or older and live in Scotland to argue that their results are
---

	representative. (Or, at least, they must describe that they will conduct analyses to check whether their sample is representative to the target population.) 2. (Page 10-11) Concerns about potential biases depending on data collection periods. The COVID-19 pandemic was unique in the way that no one had great understanding of it. Governmental and social reaction and responses varied significantly depending on periods. The responses to the fear measures could be significantly different depending on “when” the survey was conducted. The authors described that the data were collected between 12 and 15 March 2021 and between 4 and 7 May 2021. It would be better to use the same items and show the robust results in both phases. Are the results robust in two phases? If not, is it because of the sample selection (age group differences) or the time trends? 3. Sampling for Phase 2A (Page 11) 3-1. Explain how to replicate the HAGIS and Generation Scotland samples. 3-2. Why is the expected response rate so low? 3-3. Describe the authors’ plan about how to handle differences in responses across the modes of interviews (phone, postal, vs. online) if they exist. Or, show your plan to show the robustness of your results across the modes of interviews. (Page 12) 3-4. Online sample introduces biases, especially for older adults. I am not sure if it’ the same for people in Scotland, but older Americans who have access to the internet are younger, more educated, richer, and live in urban areas. How to handle this sampling biases? Online panelists do not solve the issues of sampling biases. 4. Concerns about Phase2B I do not see much value of conducting qualitative research in the phase 2B. The quantitative data analyses will answer the most of the research questions the authors have. Is it necessary? Isn’t it better to use a longitudinal study and conduct panel analyses? I think this research will provide better implications by conducting panel analyses because of the frequent changes in policies and situations during COVID-19. Minor comments: 1. The authors still use the term denoting “causal relationships,” not “correlational relationships” throughout the manuscript. Please correct them. For example, the highlighted words below indicate causal relationship. (e.g., impact, affect, etc.) (Page 2) Lines 20-21: (2) examine the impact of COVID-19 fear (Page 6) Lines 47-48: How it impacts (Page 7) Lines 7-8: To examine the impact Line 22-23: how has COVID-19 fear impacted Lines 47-48: How has COVID-19 fear affected 2. Make sure to double check whether the authors replaced older people with older adults. I still find a couple of them. 3. (Page 5) Lines 29-31: The authors wrote “...has prompted stringent government regulations
--	---

	seeking to protect this population [1]. The UK Government issued guidance to safeguard vulnerable people during the COVID-19 pandemic ...”: I guess the authors wanted to say governmental health guidelines on social distancing. However, it is not specified. It would be better to provide a couple of examples using a parenthesis. (e.g., stay-at-home orders, etc.) 4. (Page 13) Lines 9-28: The authors must mention that their survey will include the fear instruments they developed.
--	--

VERSION 2 – AUTHOR RESPONSE

Responses to Reviewer 2 Comments	
Reviewer 2 comment	Author response
(Page 10) Lines 9-40: The authors describe their sampling method to develop the fear instrument. Their target sample of the entire project is adults aged 50 or older. However, the research used online panel 1) those who aged > 18 in the first stage and 2) those who aged > 40 in the second stage. Are these relevant? Do you have rationales for this age group? Also, it seems like the authors used online panelists who are used to participate in online surveys. Are they representative to the national sample? They must show that their sample characteristics are similar to those of people aged 50 or older and live in Scotland to argue that their results are representative. (Or, at least, they must describe that they will conduct analyses to check whether their sample is representative to the target population.)	In the scale development work, the rationales were pragmatic for (1) recruiting from Prolific.co, for (2) recruiting UK residents outside of Scotland and for (3) recruiting those under age 50. The reviewer is correct that in an ideal world we would have recruited a representative sample of older people living in Scotland. To do so in our scale development sub-project, however, would have proven prohibitively costly. In terms of time, it would have made it far harder to find our sample; relative to all adults in the UK, the number of older adults living in Scotland is quite small. In terms of money, it is far less costly to recruit from a self-selected sampling pool like Prolific than it is to recruit from a commercial panel that seeks to be representative of the population of older Scottish adults; as a side-note, the commercial panels that are available to researchers (e.g. Yougov) use a quota sample and so are representative only in terms of observable characteristics. Even if we had gone this route we would still face the problem that our sample may not be representative of the population in terms of unobservables (e.g. the quota sample might be more digitally literate than the wider population; our final survey got around this problem by recruiting by modes other than through an online survey). Our project faced time constraints - a goal of the project was to provide timely results to policy makers. Factor analysis required a sample of hundreds of respondents in each stage of data collection for these scale development surveys. We could fulfil that requirement more rapidly by expanding our sample in to comprise individuals younger than our target population and by including respondents from elsewhere in the UK other than Scotland. Given that our project faced budget constraints we had to trade off data collection for this scale development sub-project against reducing the resources for data collection for the final survey. When we came to this decision, the marginal benefit of allocating resources to the final survey were obviously high e.g. greater resources to follow up with additional respondents to the first wave of HAGIS and hence pursue the longitudinal dimension of our study. The benefit of

	allocating budget to recruiting a more representative sample for this scale development subproject did not appear to be large. What we were trying to achieve in this scale development subproject was to identify the latent factors that characterise Covid fear among older people living in Scotland and to identify which scale items do a good job of predicting those latent factors. We took a considered judgment that the latent factors that characterise Covid fear among older people living in Scotland would be similar to the latent factors that characterise Covid fear among the broader sample recruited in these scale development tests. The second round of data collection for the scale development supported this view - when we recruited a sample of those aged over 40, the factor analysis replicated the findings from the wider sample used in Stage 1. Ultimately, we could test this assumption in the data. When we run the factor analysis on the data returned by our final sample of older people living in Scotland, the alphas for each of the five amply satisfy the criterion set by Nunnally & Bernstein (1994) of exceeding 0.7; in fact, the lowest was 0.818. We add to the protocol text that we will test in the final dataset the internal consistency of our factors. Reference: Nunnally, J. C., & Bernstein, I. H. (1994). Psychometric Theory (3rd ed.). New York, NY: McGraw-Hill.
(Page 10-11) Concerns about potential biases depending on data collection periods. The COVID-19 pandemic was unique in the way that no one had great understanding of it. Governmental and social reaction and responses varied significantly depending on periods. The responses to the fear measures could be significantly different depending on “when” the survey was conducted. The authors described that the data were collected between 12 and 15 March 2021 and between 4 and 7 May 2021. It would be better to use the same items and show the robust results in both phases. Are the results robust in two phases? If not, is it because of the sample selection (age group differences) or the time trends?	We agree with Reviewer 2; the situation was evolving rapidly over the months that we were devising our scale. In addition to the factors pointed to by the Reviewer, another development was that the research literature relevant to our scale development was growing. We updated our scale items to take account of these developments. Between March (the first round of data collection) and May (and second) we encountered a scale in Mertens (2021) that contained an item that captured a concern that was missed by the items in our first-round survey: “I am worried that the coronavirus will mutate into a deadlier strain or never disappear from the population”. Specifically, we added two scale items, which we tested in our second round of factor analysis (May 4-7): “I am worried that the Covid-19 virus will mutate into a deadlier strain” and “I am worried that the Covid-19 virus will never disappear from the population”.

	These two items robustly loaded onto a factor that had been identified in the first round of data collection – long term concerns. The factor structure recommended by the first round of data collection was confirmed in the second round by both exploratory and confirmatory factor analysis. In short, the first and second rounds of data collection returned results that converged in which factors they recommended. The scale items differed across the first- and second round of data collection because the second round included two items that did a better job than previous items of capturing one of those factors (Long term concerns). These results are reported more fully in our paper on scale development (Comerford et al., 2022). References: Comerford, D. A., Olivarius, O., Bell, D., & Douglas, E. (2022). Validation of the Worries Emerging from the Covid-19 Pandemic (WECP) Scale. Mertens, G., Duijndam, S., Smeets, T., & Lodder, P. (2021). The latent and item structure of COVID-19 fear: A comparison of four COVID-19 fear questionnaires using SEM and network analyses. Journal of Anxiety Disorders, 81, 102415.
Sampling for Phase 2A (Page 11) 3-1. Explain how to replicate the HAGIS and Generation Scotland samples.	The criteria for inclusion in the drawn sample from both HAGIS and Generation Scotland were people living in Scotland who were aged 50 and over at the time of the survey.
3-2. Why is the expected response rate so low?	Our expected response rate of 25% was based on previous research suggesting that response rates have varied over the period of the COVID-19 pandemic with some evidence of survey fatigue. This evidence is cited on page 11.

	Reference: Fawns-Ritchie, C., Altschul, D. M., Campbell, A., Huggins, C., Nangle, C., Dawson, R., ... & Porteous, D. J. (2021). CovidLife: a resource to understand mental health, well-being and behaviour during the COVID-19 pandemic in the UK. Wellcome Open Research, 6(176), 176.
3-3. Describe the authors' plan about how to handle differences in responses across the modes of interviews (phone, postal, vs. online) if they exist. Or, show your plan to show the robustness of your results across the modes of interviews.	We intentionally chose to adopt different modes to promote inclusivity of responses to the survey. For example, to reach people without internet access. We recorded the mode for all participants and have undertaken initial analyses to test for mode effects. Further details of these will be developed to inform future waves of the study, and where relevant, for publication. Users of the data will be able to control for/include/exclude different modes from their analyses.
(Page 12) 3-4. Online sample introduces biases, especially for older adults. I am not sure if it's the same for people in Scotland, but older Americans who have access to the internet are younger, more educated, richer, and live in urban areas. How to handle this sampling biases? Online panelists do not solve the issues of sampling biases.	Thank you for this input. Our use of different modes was our strategy to overcome these biases. We recontacted people using their preferred mode of contact which included postal and telephone. Further, we are computing and including sample weights that account for under-/over-representation of specific sub-groups.
Concerns about Phase2B I do not see much value of conducting qualitative research in the phase 2B. The quantitative data analyses will answer the most of the research questions the authors have. Is it necessary? Isn't it better to use a longitudinal study and conduct panel analyses? I think this research will provide better implications by conducting panel analyses because of the frequent changes in policies and situations during COVID-19	We have a strong belief that the qualitative strand of our work helps to develop a deep and rich understanding of older adults' COVID-19 fears and worries, and pandemic-related experiences and behaviours. Indeed, the qualitative outputs have helped to inform the interpretation of the quantitative responses in the survey.
The authors still use the term denoting "causal relationships," not "correlational relationships" throughout the manuscript. Please correct them. For example, the	We believe the terms impact and effect do not indicate causal relationships. We did not use the term causes anywhere in the manuscript. Further, in our

highlighted words below indicate causal relationship. (e.g., impact, affect, etc.) (Page 2) Lines 20-21: (2) examine the impact of COVID-19 fear (Page 6) Lines 47-48: How it impacts (Page 7) Lines 7-8: To examine the impact Line 22-23: how has COVID-19 fear impacted Lines 47-48: How has COVID-19 fear affected	data analysis plan for Phase 2, we made it clear that our intentions are to identify correlates of COVID-19 fear and test associations.
Make sure to double check whether the authors replaced older people with older adults. I still find a couple of them.	We thank Reviewer 2 for this observation. We have now replaced all occurrences of older people with older adults.
(Page 5) Lines 29-31: The authors wrote "...has prompted stringent government regulations seeking to protect this population [1]. The UK Government issued guidance to safeguard vulnerable people during the COVID-19 pandemic ...": I guess the authors wanted to say governmental health guidelines on social distancing. However, it is not specified. It would be better to provide a couple of examples using a parenthesis. (e.g., stay-at-home orders, etc.)	We have now added examples to clarify what we meant by the UK Government issued guidance. Please refer to page 5 to see changes which read: The UK Government issued guidance (e.g., stay-at-home orders, shielding, social distancing) to safeguard vulnerable people during the COVID-19 pandemic.
(Page 13) Lines 9-28: The authors must mention that their survey will include the fear instruments they developed.	We indicated in our manuscript that the survey will include the Worries Emerging from the COVID-19 Pandemic (WECP) instrument. Please refer to page 13 to read: The WECP scale will be incorporated into the survey instrument.

VERSION 3 – REVIEW

REVIEWER	Shin, Su University of Utah Health
REVIEW RETURNED	01-Jan-2023
GENERAL COMMENTS	NA